# Investigation of Impact Resistance of High-Strength Portland Cement Concrete Containing Steel Fibers

**DOI:** 10.3390/ma15207157

**Published:** 2022-10-14

**Authors:** Mohammad Mohtasham Moein, Ashkan Saradar, Komeil Rahmati, Arman Hatami Shirkouh, Iman Sadrinejad, Vartenie Aramali, Moses Karakouzian

**Affiliations:** 1Department of Civil Engineering, Allameh Mohaddes Nouri University, Nour 4641859558, Iran; 2Department of Civil Engineering, University of Guilan, Rasht 419961377, Iran; 3Department of Civil Engineering, Somesara Branch, Islamic Azad University, Somesara 4361947496, Iran; 4Department of Civil Engineering, Qazvin Branch, Islamic Azad University, Qazvin 1519534199, Iran; 5Department of Civil Engineering, Ramsar Branch, Islamic Azad University, Ramsar 4487136889, Iran; 6Department of Civil Engineering and Construction Management, California State University, Northridge, LA 91330, USA; 7Department of Civil and Environmental Engineering and Construction, University of Nevada, Las Vegas, NV 89154, USA

**Keywords:** high-strength concrete, fiber-reinforced concrete, steel fibers, impact strength, curing conditions

## Abstract

Impact resistance of Portland cement concrete (PCC) is an essential property in various applications of PCC, such as industrial floors, hydraulic structures, and explosion-proof structures. Steel-fiber-fortified high-strength concrete testing was completed using a drop-weight impact assessment for impact strength. One mix was used to manufacture 320 concrete disc specimens cured in both humid and dry conditions. In addition, 30 cubic and 30 cylindrical specimens were used to evaluate the compressive and indirect tensile strengths. Steel fibers with hooked ends of lengths of 20, 30, and 50 mm were used in the concrete mixtures. Data on material strength were collected from impact testing, including the number of post-first-crack blows (INPBs), first-crack strength, and failure strength. Findings from the results concluded that all the steel fibers improved the mechanical properties of concrete. However, hooked steel fibers were more effective than crimped steel fibers in increasing impact strength, even with a smaller length-to-diameter ratio. Concrete samples containing hybrid fibers (hooked + crimped) also had lower compressive strength than the other fibers. Comparisons and analogies drawn between the test results and the static analyses (Kolmogorov–Smirnov and Kruskal–Wallis) show that the *p*-value of the analyses indicates a more normal distribution for curing in a humid environment. A significant difference was also observed between the energy absorptions of the reinforced mixtures into steel fibers.

## 1. Introduction

Concrete is the most used building material, with an annual consumption of well over 25 billion tons [1,2,3,4,5]. The availability of ingredients, ease of casting in any shape, economy, high compressive strength, as well as good durability in various environmental conditions are the factors that make concrete such a versatile construction material [6,7,8,9]. The use of concrete in constructing different structures, such as bridges, high-rise buildings, dams, and nuclear power plants, is inevitable, and as such, the concrete industry has seen remarkable progress in recent years. This progress has led to the development and production of high-strength concrete (HSC) and high-performance concrete (HPC) with desirable properties [10,11]. High-strength concrete has acceptable performance under harsh climates and humid environments due to its low permeability properties, thus decreasing maintenance and repair expenses [12,13,14]. Plain concrete, however, is generally fragile [13], with low tensile strength [15]. Different fibers are often added to concrete to improve the tensile strength, ductility, and toughness properties [1,16]. Generally, fibers can be made from manufactured products such as steel, glass, and synthetics or natural materials such as hemp, jute, and silk [15]. Among the various fibers used in manufacturing fiber-reinforced concrete, significant attention has been devoted to the use of steel fibers. Studies show that the presence of steel fibers can improve the properties of concrete, including increased compressive and tensile strength [17,18,19], thereby increasing energy absorption [17] as well as reducing water absorption by controlling cracks and mitigating them [19].

Concrete impact resistance is an essential property in various applications such as industrial floors, hydraulic structures, and explosion-proof structures. Given that the use of fiber-reinforced concrete in these structures is increasing, it is important to evaluate the impact resistance of concrete using different types of fibers. Among the various methods for assessing the impact strength of concrete, the drop-weight impact test proposed by ACI Committee 544 [20] is the simplest method. In this method, the impact strength of the concrete is evaluated based on the number of blows needed to produce the first crack and failure.

Some factors, such as the test character or the heterogeneous nature of concrete, can cause the impact test results to be highly dispersed [21]. Various studies have, therefore, investigated the statistical and experimental properties of fiber-reinforced concrete’s impact strength using the ACI 544 method. This study evaluates the effects of different curing conditions and different types of fibers, namely, hook and crimped steel fibers, and the impact resistance of fiber-reinforced concrete. Fiber-reinforced concrete mix designs were made and cured in either humid or dry conditions. Then, the drop-weight impact test was conducted, and impact strength parameters were determined for different curing conditions and different types of steel fiber.

## 2. Research Background

Many researchers have studied the effect of steel fibers on the mechanical properties of different types of concrete. A complete report on the characteristics of consumable steel fibers and the types of tests performed in previous studies is reported in Table 1.

According to Table 1, about 54 papers containing useful information about concrete containing steel fibers were reviewed. The results indicate that investigating the effect of different shapes of steel fibers in concrete and comparing them can be considered for future studies. On the other hand, the length/diameter parameter is one of the most important items regarding steel fibers. According to the investigations, few papers have investigated the different length/diameter ratios in the laboratory and numerical field. The concrete curing environment is also one of the items affecting the mechanical characteristics of concrete, but most of the studies conducted on concrete containing steel fibers against impact strength are ultimately limited to a curing environment. Figure 1 shows the overview of the study process. In the present study, steel fibers are used as an effective factor in increasing the mechanical strength of concrete. Although various studies have been conducted on concrete containing steel fibers, the study of this type of concrete against impact load requires further investigation. Factors such as specimen geometry, fiber type, concrete mixture, nature of the test, and the nonhomogeneous condition of the concrete are very effective in the weight drop test [21,26]. Therefore, the statistical and laboratory investigations of the impact resistance of concrete containing steel fibers can provide clearer performance results for this type of concrete.

Therefore, in this study, in the first stage, a complete study was performed on the research on fiber-containing concrete and information such as concrete type, the shape of steel fibers, fiber length, fiber diameter, the length-to-diameter ratio of fibers, and fiber consumption dose; the mechanical tests performed on concrete are reported in a table. In the next step, to study the effect of the shape and length of fibers, steel fibers in different sizes were used. Additionally, to investigate the effect of curing conditions on the mechanical properties of concrete, two completely different curing environments were planned. After the fabrication and processing of the samples, compressive, tensile, and impact strength tests were used to evaluate mechanical strength. In order to provide clearer results, the results of the experiments were analyzed by statistical analysis. The behavior of concrete under short-duration dynamic loads (such as earthquakes, wind gusts, and machinery) is completely different from the behavior it exhibits under static load conditions. Therefore, the results of the present study can be considered a practical reference for projects that are of high importance, such as hospitals, nuclear power plants, tall buildings, etc. It is expected that the results of this study can provide useful information to civil engineers.

## 3. Materials and Methods

### 3.1. Materials

In our study, Portland Cement Type II (Table 2) was added to the concrete mixtures. Silica fume (Table 2) was also considered a mineral pozzolanic material to replace part of the weight of cement. For a coarse aggregate, gravel with a water absorption of 0.8% and the largest nominal grain size of 19 mm was used. For the fine aggregate component, natural sand with a water absorption of 1.3% and a maximum diameter of 4.75 was used. The particle size distributions of the fine and coarse aggregates are shown in Figure 2. The superplasticizer (Table 3) employed in the present study was derived from a basis of polycarboxylate with a specific weight of 1.2 ± 0.05 gr/cm^3^. The steel fibers utilized during this investigation had a tensile strength of 1140 MPa and were crimped and hooked (Table 4, Figure 3).

### 3.2. Mixing of Concrete and Casting of the Test Specimens

One concrete mixture with a water-to-binder (W/B) ratio of 0.4 and a cement grade of 450 kg/m^3^ was designed. The concrete mix proportions are given in Table 5. The concrete mixtures consisted of a control mix with no fiber and four combinations with fibers. The first mixture is called control concrete (R), which has no fibers. In the second mixture (F2), 2 cm crimped fibers were used to make the fiber concrete. Additionally, 3 cm hooked fibers and 5 cm crimped fibers were added to the third (F3) and fourth (F5) mixtures, respectively. Finally, in the fifth blend (F2F3), a combination of 2 cm crimped steel fibers and 3 cm hooked steel fibers was used. The distribution of steel fibers in all the fiber concrete mixtures was 1% by volume [34,71].

Figure 4 shows the results of the slump test of mixtures. For concrete without fibers, the amount of slump was 19 cm, while for other mixes that had steel fibers, a lower amount of slump was obtained. The results of the slump test of the mixtures show that the inclusion of steel fibers reduces the workability of the concrete. The result is confirmed by the study of Atis and Karahan [35].

After conducting the slump test, the concrete mixture was poured into 15 cm cubic molds for the compressive strength test and 30 × 15 cm cylindrical molds for the splitting tensile strength test (Figure 5). After 24 h, the samples were removed from the mold. One set of specimens was cured in humid conditions and another set was cured in a dry environment until they were ready to be tested. According to the ACI 308R-01 standard [72], in order to process the samples in a humid environment, the samples were placed in water at 23 ± 1 °C for 28 days; in order to dry the samples in a dry environment, the samples were placed in a laboratory environment at 25 °C for a 28-day curing period. The number of days was set.

The impact test was conducted in line with the drop-weight measurement recommended by ACI Group 544. As shown in Figure 6, this test consists of repeatedly dropping a hammer weighing 4.54 kg from a height of 457 mm on a steel ball with a radius of 63.5 mm located at the center of the sample’s surface. The strength of the initial crack was obtained by measuring the number of blows that resulted in the initial crack. The failure strength refers to the number of blows required to touch three of the four metal bushes of the impact device by the concrete disc. This test was performed on each type of concrete at 28 days of age on 64 concrete discs (32 disks in a humid environment and 32 disks in a dry environment) with a diameter of 15 cm and a thickness of 6.4 cm, having been cut from 15 × 30 cm cylindrical specimens (Figure 7). Details of the number of specimens and discs used for impact testing are reported in Figure 8.

### 3.3. Tests

The tests were performed on hardened concrete after a curing period of 28 days. The test specification, the size of the samples, and the standards used are given in Table 6. Moreover, in order to examine the efficiency of the fresh concrete, the slump test was performed on the fresh concrete. Figure 9. shows an overview of the experiments performed in the study. Repeatability (or reliability of re-testing) is a parameter used to describe a change in successive measurements of the same variable under similar conditions over a short period of time. The repeatability index is defined as follows [73,74]:(1)$$\mathrm{RI}=\frac{\Sigma CV\%}{N_{all}}$$
where Σ*CV% =* is the sum of coefficients of variation percentages and *N_all_* = is the total number of mixtures.

Table 7 shows the results of repeatability testing for previous studies and the current study. Comparing the result indicator RI obtained for the present study with the previous studies shows that the results of the tests are stable and repeatable.

## 4. Results and Discussion

### 4.1. Compressive Strength

The 28-day specimens’ compressive strength was determined for humid and dry curing conditions, as shown in Figure 10a. The compressive strength of the specimens varied from 46 to 70 MPa. Figure 10a shows that, regardless of the type of curing, the addition of steel fibers increased the compressive strength of the high-strength concrete. On the other hand, the results showed that using crimped fibers with a longer length-to-diameter ratio resulted in higher compressive strength. Curing the specimens in a humid environment also resulted in higher compressive strength than those cured in a dry environment. This is attributed to the higher degree of hydration. This result was also reported by Bingöl and Tohumcu [82], Razzaghi et al. [83], and Benli et al. [84].

Examining fiber performance in concrete shows that 5 cm corrugated fibers increased the compressive strength of ordinary concrete by an average of 39%, while 3 cm and 2 cm corrugated fibers improved the compressive strength by 33% and 28%, respectively. A comparison of the mixtures’ compressive strengths indicates that the mixture containing 5 cm crimped fibers was 10%, 5%, and 9% stronger, on average, than F2, F3, and F3F2. This may be due to the longer length of the 5 cm wavy fibers compared to other fibers.

### 4.2. Splitting Tensile Strength

Figure 4b shows the 28-day splitting tensile strength of different concrete samples under both humid and dry curing conditions. The lowest splitting tensile strength was 3.15 MPa for the control concrete and concrete cured in dry conditions. This value increased up to 5.2 MPa for steel fibers and concrete cured in a humid environment. These findings reflect the fact that the tensile strength of the high-strength concrete showed increased values when using steel fibers. On the other hand, the results of Figure 10b show that the 3 cm hooked steel fibers resulted in higher tensile strength than the other fibers. The positive effect of curing in a humid environment compared to a dry environment is displayed in Figure 10b. The studies of Benli et al. [84], Haghighatnejad et al. [85], and Ismail et al. [86] also confirm the result. The addition of 3 cm hook fibers improved the tensile strength of the reference concrete by 19%, whereas 5 and 2 cm crimped fibers provided 8% and 14% improvement in the reference concrete, respectively.

### 4.3. Impact Strength

For each mixture and curing condition, 32 discs were tested for the drop-weight impact load. The number of blows required for the appearance of the first crack and for the final failure was determined in the tests. The results for the humid and dry curing conditions are given in Table 8. Regarding the statistical study of initial crack strength, fracture strength, and the increase in the number of post-first-crack blows (INPBs), two statistical procedures were chosen: the Kolmogorov–Smirnov and Kruskal–Wallis analyses. The Kolmogorov–Smirnov test examines the probability of whether the observations are normal or not. In this test, a *p*-value less than 0.05 means that the normal assumption was rejected. On the other hand, the Kruskal–Wallis test compares two or more groups of data. In the Kruskal–Wallis test, there was no significant difference between groups, as the *p*-value was greater than 0.5 [21].

#### 4.3.1. First Crack Strength

Table 8 shows that the average number of blows needed for the appearance of the first crack was 144 in high-strength control concrete (R) cured in a humid environment. For the same curing conditions, the average number of blows for the first crack of fiber-reinforced concrete specimens of mixtures F2, F3, F5, and F2F3 were 172, 416, 223, and 173, respectively (Figure 11a). The presence of steel fibers in the concrete, therefore, enhanced the number of blows needed for the first crack. This is because the steel fibers reinforce the concrete in three directions, which helps the disk to absorb impact energy. The increased energy absorption capacity could eventually delay the emergence of the first crack [26]. Aghaei et al. [87] also reported an improvement in the concrete’s impact strength by adding steel fibers.

The results given in Table 8 show that high-strength concrete containing 5 cm crimped steel fibers can withstand a higher number of average blows than concrete containing 2 cm crimped steel fibers. This shows the positive effect that fibers with a longer length-to-diameter ratio have on impact strength. Fiber shape, however, also plays a role. The hooked fibers increased the impact strength of the high-strength concrete better than the crimped fibers did, even with a stronger length-to-diameter ratio. The trend observed for the performance of steel fibers in a humid environment is similar to that for curing in dry environments. In dry curing conditions, an average of 123 blows was required to create the first crack in the high-strength control concrete, which increased to 147, 227, 158, and 144 in the fiber-reinforced concretes F2, F3, F5, and F2F3, respectively (Figure 11a). For the mixtures, a 95% confidence interval was obtained on the average resistance of the first crack, which includes the lower and upper bounds (Table 8). This range indicates that there is a 95% chance that the true first-crack resistance is within this range [21,26].

The coefficients of the variation of the first-crack impact strength for mixtures R, F2, F3, F5, and F2F3 in a humid environment were 27%, 33%, 77%, 36%, and 37%, respectively. The corresponding values for dry curing conditions were 29%, 32%, 88%, 41% and 44%, respectively. Thus, the impact strength data shows more dispersion for dry-cured concrete than those cured in humid conditions.

As stated, *p*-values of less than 0.05, obtained by the Kolmogorov–Smirnov test, indicate that the data are not normally distributed. The normal probability shapes and histograms for different mixtures cured in humid conditions are plotted in Figure 12, Figure 13, Figure 14 and Figure 15. Figure 16, Figure 17, Figure 18 and Figure 19 show the corresponding values for dry curing conditions.

Figure 12a shows the frequency histograms, along with the normal plot, for reference concrete processed in a wet environment, which means that the resistance of the first crack of the reference concrete rarely follows a normal distribution. In addition, Figure 13a shows the normal probability plot of this mix. There is a low curvature plot in the lower half of the straight line that indicates a rarely normal distribution. On the other hand, the normal distribution of results was examined by the Kolmogorov–Smirnov test with a significance level of 0.05 (reported in Table 8). The *p*-value obtained for the first cracks of the reference concrete in a wet environment was 0.093, which confirms the above claim. Mixtures of F2 and F23 follow the normal distribution in Figure 13b and Figure 15b, respectively, due to the relatively close fit of the normal plot to the histogram (Figure 12b and Figure 14b, respectively) and the almost linear pattern of the data. Additionally, the *p*-value obtained by the K–S test showed a value greater than 0.15, which confirms their normal distribution. Mixtures of F3 and F5, however, do not follow the normal distribution due to the normal plot’s poor fit, shown in Figure 12c and Figure 14a, respectively, aside from the intensive deviation of the results from the straightforward line (Figure 13c and Figure 15a). The *p*-value of the K–S test was less than 0.01. Figure 16a and Figure 17a show the normal distribution of reference concrete in a dry environment due to the acceptable fit of the normal plots to the histogram and linear pattern of the data. On the other hand, this mixture’s *p*-value was equal to 0.141. Mixtures of F3 and F2F3 do not follow the normal distribution due to the poor representation in Figure 16c and Figure 18b as well as the nonlinear pattern in Figure 17c and Figure 19b. In addition, the *p*-value confirms this result from the K–S test. Finally, the incorporation of F5, according to Figure 18a and Figure 19a, as well as a *p*-value equal to 0.093, shows a rarely normal distribution. Table 8 confirms that the first-crack impact strengths of concretes F2 and F2F3 cured in a humid environment and those of mixtures R and F2 cured in a dry environment have shown approximately normal distributions.

Table 9 illustrates the comparison of the first-crack strength in the two curing environments against conventional high-strength concrete using the Kruskal–Wallis test. The results show that the first-crack impact strength in a humid environment for mixtures F2, F3, F5, and F2F3 compared to the reference high-strength concrete has *p*-values of 0.029, 0.000, 0.000, and 0.049, respectively. In addition, the corresponding *p*-values for dry curing conditions were 0.030, 0.001, 0.021 and 0.321, respectively. This indicates a significant improvement in the first-crack impact strength of high-strength concrete when the steel fibers were introduced separately compared to that of the high-strength control concrete.

#### 4.3.2. Failure Strength

According to the average strength values presented in Table 8, in humid environments, the average failure strength of conventional high-strength concrete is 150 blows. The average failure strength of F2, F3, F5, and F2F3 mixtures are 192, 455, 257, and 200 blows, respectively, indicating the improved failure strength of high-strength concrete with the addition of steel fibers (Figure 11b).

Similarities were observed in the strength of the first crack, the mean blows’ strength at failure, and the high-strength concrete containing 5 cm crimped steel fibers increasing from 2 cm crimped steel fibers. This demonstrates the effective role of a higher length/diameter ratio in increasing the impact strength of the concrete. Hooked fibers, however, have better performance in increasing the concrete’s strength compared to crimped fibers, even with a lower diameter ratio. In a dry environment, the average failure strength of conventional high-strength concrete is 129 blows, and the mean failure strength of F2, F3, F5, and F2F3 mixtures is 158, 237, 173, and 163, respectively (Figure 11b). As with the first-crack strength, a 95% confidence interval on the mean for failure strength was obtained (Table 8). This range indicates that there is a 95% chance that the true failure strength is within this range [21,26].

Therefore, in dry environments, as in humid environments, the addition of steel fibers to high-strength concrete improves failure strength. Average impacts for failure in humid environments compared to dry environments for conventional high-strength concrete were 1.162 times, for concrete containing 2 cm crimped steel fiber 1.215 times, for 3 cm hooked steel fiber 1.919 times, for 5 cm crimped steel fiber 1.485 times, and for fiber composition 1.226 times. Similar to the strength of the first crack, the comparison of the failure strength coefficient of variation in the two environments also shows higher values in dry environments than in humid environments.

Figure 20, Figure 21, Figure 22 and Figure 23 show the histogram shapes and the normal probability of failure strength for humid environments. Additionally, Figure 24, Figure 25, Figure 26 and Figure 27 show the same for failure strength in dry environments. The shapes and results of the Kolmogorov–Smirnov test in Table 8 show the F5, F2, and F2F3 mixtures’ almost normal distribution of failure strength in a humid environment. The distribution of the F2 mixture was also almost normal in a dry environment.

Figure 20a shows the frequency histogram with the normal plot, and Figure 21a shows the normal probability plot for the failure resistance R. Additionally, the *p*-value of the K–S test, which is equal to 0.068, is the result of a seldom normal distribution for this mixture. Figure 20b and Figure 22a,b, which are related to mixtures of F2, F5, and F2F3, respectively, show a relatively close fit of the normal plot to the histogram. In addition, Figure 21b and Figure 23a,b demonstrate the almost linear pattern of the mixed data. The *p*-value for mixtures F2, F5, and F2F3 by the K–S test indicates a number greater than 0.15, which confirms the normal distribution of these mixtures. For the F3 mixture, Figure 20c and Figure 21c, plus the *p*-value (0.087), show that a normal distribution is rare. Reference concrete in a dry environment also rarely shows a normal distribution due to the poor display in Figure 24a and Figure 25a. The *p*-value for this mixture is equal to 0.075, which confirms its rarely normal distribution. The F2 mixture shows a normal distribution due to the relatively acceptable results of the normal plot with the histogram in Figure 24b, the almost linear pattern of the data in Figure 25b, as well as a *p*-value greater than 0.15 from the K–S test. Mixtures of F3, F5, and F2F3 do not follow the normal distribution due to poor representation of the fit between the normal plot and the histogram in Figure 21c and Figure 22a,b, respectively. This is also due to the nonlinear pattern in Figure 25c and Figure 27a,b, respectively. The *p*-value obtained from the K–S test for these mixtures also indicates abnormal distributions.

Table 9 compares the failure strength of two curing environments with the Kruskal–Wallis test against conventional high-strength concrete. The results show that the failure strength of humid environments for F2, F3, F5, and F2F3, with respect to conventional high-strength concrete, have *p*-values of 0.001, 0.000, 0.000, and 0.001, respectively. In dry environments, the values obtained were 0.003, 0.000, 0.002, and 0.058, respectively.

Figure 28 displays the failure pattern of the control concrete (no steel fibers) and fiber sample concrete under the drop-weight impact test. As expected, non-steel discs exhibited brittle behavior and failed to gain centralized, sudden, and massive cracking (Figure 28a). Additionally, fewer blows were recorded from the time of the first crack to final disc failure. Examination of the crack pattern in the discs containing steel fibers showed multiple and interconnected cracks (Figure 28b). In addition, the separation of particles in the discs and the release of fibers in these types of mixtures were accompanied by the maintenance of matrix integrity. This is due to the bridging properties of the fibers, which can be seen in Figure 28c.

#### 4.3.3. The Frequency of Blows after the Initial Crack

The comparison in Table 8 shows that in humid and dry environments, the average number of blows after the first crack until failure for the high-strength control concrete was six blows. The values for mixtures F2, F3, F5, and F2F3, cured in a humid environment, were 6.5, 3.33, 5.66, and 4.5 times that of the control concrete, respectively. Similarly, the average number of blows after the first crack until failure for mixtures F2, F3, F5, and F2F3, cured in a dry environment, was 1.83, 1.83, 2.5, and 3.16 times that for the control concrete, respectively. This again demonstrates the efficiency of reinforced high-strength concrete with steel fibers. These figures also show that curing in humid environments can improve the number of blows after the first crack in fiber concrete. Table 9 shows the comparison of the INPB in the two curing environments through the Kruskal–Wallis test against the high-strength control concrete. The results show that concrete containing steel fibers in both curing environments significantly increases the frequency of blows following the occurrence of the first crack, which means that the fibers inhibit the propagation of cracks and delay final failure.

#### 4.3.4. Ductility Index and Impact Energy

In the field of impact resistance, in addition to first-crack strength and failure strength, a new term, the ductility index (*λ*), has been defined by Cao [88]. This index reflects the toughness of concrete discs after cracking [49]. The ductility index is defined as the ratio of the impact energy absorbed by the concrete from crack to failure (*N_fail_* − *N_first_*) to the impact energy absorbed by the first visible crack in the concrete (*N_first_*). The ductility index (*λ*) and impact energy are defined by the following formula [49,89]:(2)$$\lambda =\frac{N_{fail}-N_{first}}{N_{first}}$$
(3)$$E=N_{fail}\times mgh$$
where *λ* = the ductility index, *N_first_* = first visual crack, *N_fail_* = ultimate crack*, E* = impact energy (J), *m* = mass of hammer, *g* = gravitation acceleration, and *h* = height of drop.

The ductility index (*λ*) for different mixtures in Table 10 shows that the mixtures containing steel fibers have a better performance in increasing the toughness of concrete after cracking. This is due to the bridging property of the steel fibers, which gives the steel fibers an extraordinary ability to absorb higher impact energy after cracking [25,63,90]. The findings obtained in the present research are consistent with previous studies [21,91,92]. According to Table 10, it is observed that the addition of steel fibers significantly increases the impact energy; this process occurs both in the first-crack stage and in the failure stage [13,93]. This indicates that the presence of fibers delays the initiation of the first crack and prevents crack propagation [49]. Additionally, Figure 29 shows a schematic view of impact energy for different mixtures. Comparing the results of mixtures in both environments indicates that the lowest value of impact energy is related to mixtures without fibers (R). Additionally, the F3 mixture shows the highest amount of impact energy in both curing environments.

#### 4.3.5. Prediction of Impact Failure Strength

Prediction equations for fracture strength in humid and dry environments are shown in Table 11, where *N_p-fail_* represents the failure strength and *N_first_* the first-crack strength according to the regression analysis of the impact strength data. According to the regression analysis, the linear relationship between the strengths at the first crack and failure in concrete specimens in both environments is supported by a suitable correlation coefficient.

The coefficient of determination (R^2^) of mixtures cured in humid and dry environments is shown in Table 11. Obtaining a coefficient of determination of 0.7 or higher, for most statistical experts, results in a reasonable model [21,94]. Based on the results, the obtained equations can successfully illustrate the relationships between the first-crack strength, the failure of conventional high-strength concrete samples, and the high-strength mixtures armed with steel fibers. The results are confirmed by the studies in Table 12 about the linear relationship between the first and ultimate crack impact resistance. In order to evaluate the prediction models’ accuracy, *MAD* and *MAPE* were used [21,26,95]:(4)$$MAD=\frac{\sum \left|N_{M}-N_{P}\right|}{n}$$
(5)$$MAPE=\frac{1}{n}\;\sum \left|\frac{N_{M}-N_{P}}{N_{M}}\right|\times 100$$

*MAD* represents the mean absolute deviation, and *MAPE* measures the mean absolute percentage error, where *N_p_* is the predicted fracture strength, and *N_M_* denotes the failure strength measured. Additionally, *n* shows the number of samples tested. Table 12 shows the *MAD* and *MAPE* values derived from the forecasting models. All mixtures exhibit relatively low *MAD* and *MAPE* values in the two environments, and this may contribute to the prediction of failure strength projections with higher accuracy [49].

### 4.4. Validation of Results

Table 13 shows the results of previous studies. Items such as the type of fiber, the amount of fiber consumption, the process of compressive and tensile strength, the number of blows for the resistance of the first crack and failure, INPB, proposed equations, *MAD*, *MAPE*, R^2^, and *p* from the Kolmogorov–Smirnov test were collected. The comparison of the results of the present study with the results of previous studies is confirmed.

## 5. Conclusions

This study statistically analyzes the effect that different types of steel fibers have on the impact strength of high-strength concrete under different curing conditions, and it extracts the following results:The presence of different steel fibers improves the compressive strength by 22% to 40%, and 5 cm corrugated fibers allow the greatest increase in strength. In addition, humid-cured samples have higher compressive strength (up to 12%).The presence of different steel fibers improves the tensile strength by 17% to 60%, and 3 cm hooked-end steel fibers allow the greatest increase in strength. The humid-cured samples also have higher tensile strength (up to 21%).The presence of steel fibers in concrete improves some parameters. For example, the number of blows for first-crack strength, failure strength, and energy absorption increase by 1.194–2.288 times, 1.224–3.033 times, and 1.83–6.5 times, respectively. Fiber shape also had an impact on the strength of the concrete, with the best performance being the 3 cm hooks.The comparison of F2 and F5 shows that taking into account the ratio of length to diameter, the presence of crimped fibers with a higher ratio of length to diameter has a better performance in increasing the strength of the first crack and failure.The relationships obtained in this study indicate a high correlation between first-crack strength and failure strength. Statistical analysis (Kolmogorov–Smirnov and Kruskal–Wallis) also shows a more normal distribution of humid environments and a significant difference in steel-reinforced concrete by *p*-value.In addition to adding fibers to concrete, proper curing can also improve impact strength. This shows the importance of the hydration process for concrete.

## Figures and Tables

**Figure 1 materials-15-07157-f001:**
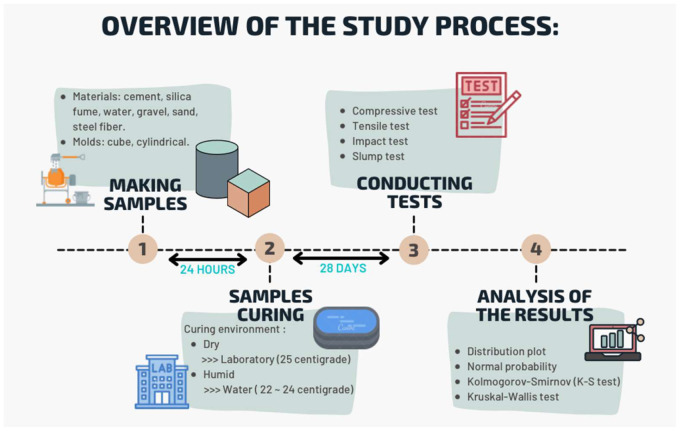
Overview of the study process.

**Figure 2 materials-15-07157-f002:**
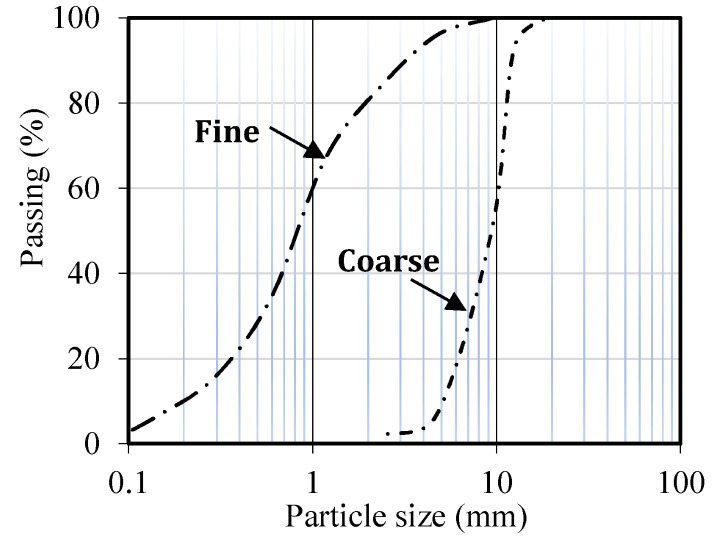
Particle size distributions of the fine and coarse aggregates.

**Figure 3 materials-15-07157-f003:**
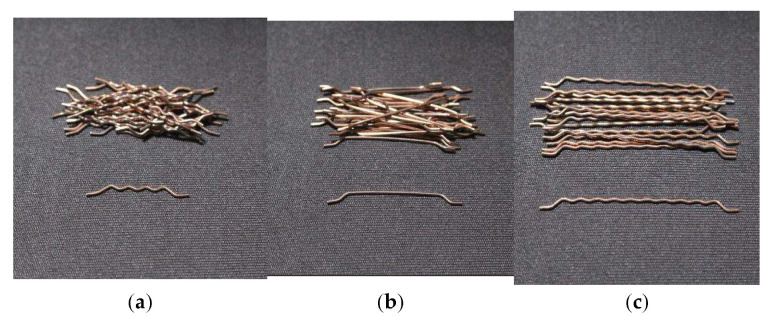
(**a**) Crimped hooked-end steel fiber [2 cm]; (**b**) Hooked-end steel fiber [3 cm]; (**c**) Crimped hooked-end steel fiber [5 cm].

**Figure 4 materials-15-07157-f004:**
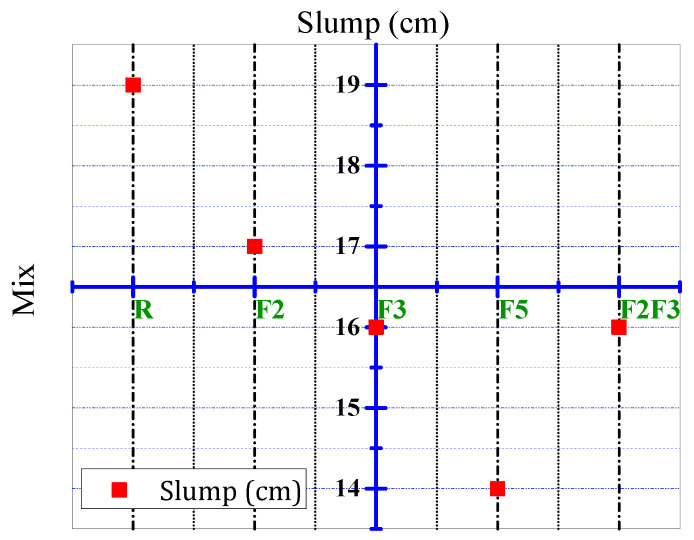
Result of slump test.

**Figure 5 materials-15-07157-f005:**
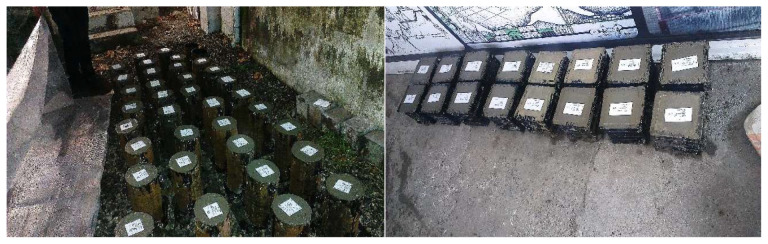
Molding of cylindrical and cubic specimens.

**Figure 6 materials-15-07157-f006:**
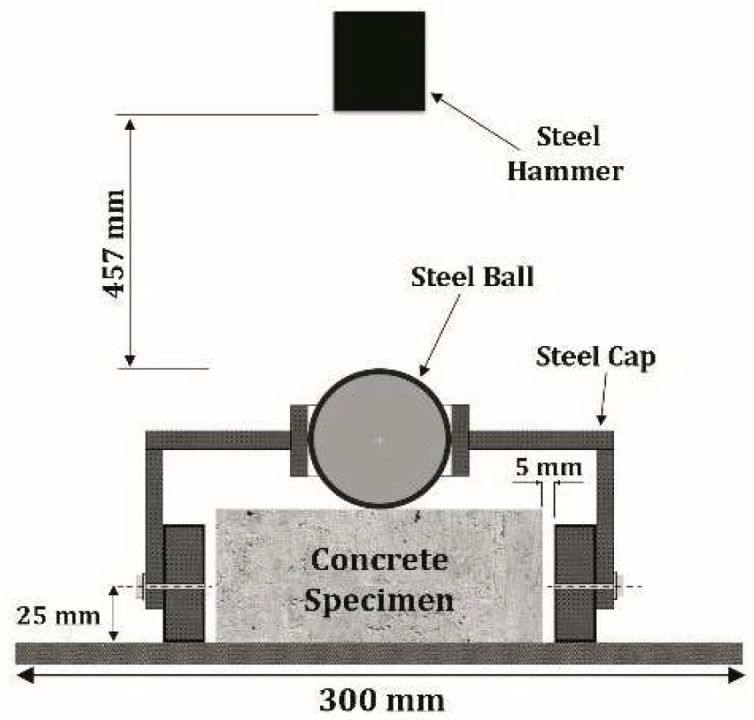
Schematic diagram of impact test—specimen setup.

**Figure 7 materials-15-07157-f007:**
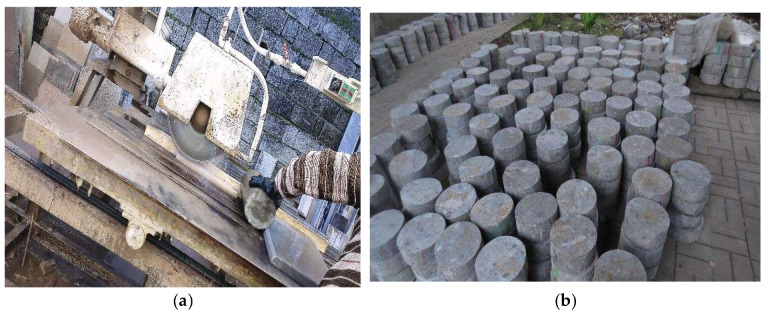
Cut samples (**a**) cutting machine (**b**) concrete disc specimens.

**Figure 8 materials-15-07157-f008:**
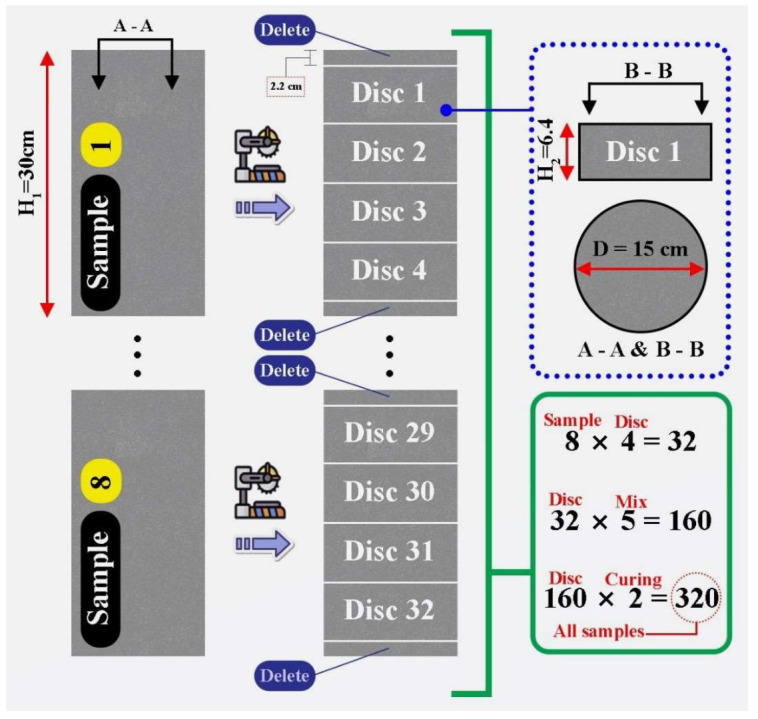
Information on samples and discs used in impact testing.

**Figure 9 materials-15-07157-f009:**
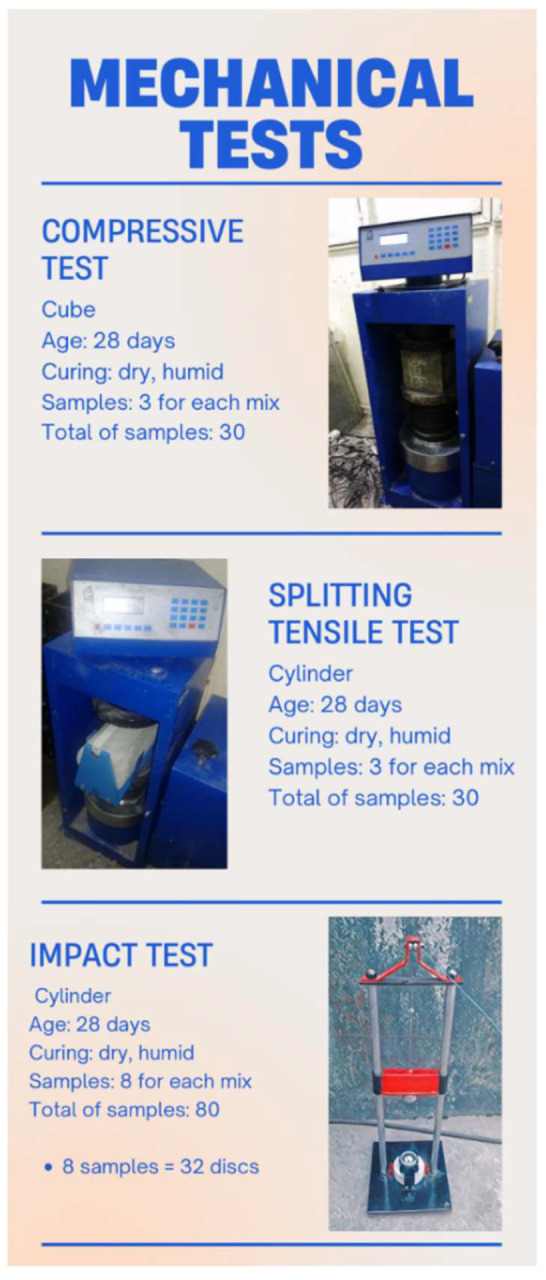
Mechanical tests performed.

**Figure 10 materials-15-07157-f010:**
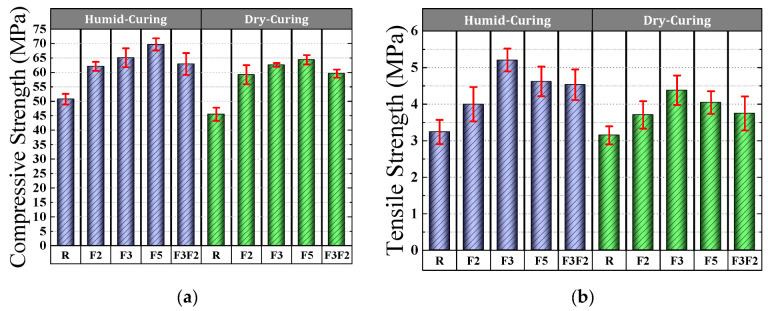
Strength of the concrete mixes: (**a**) 28 days compressive tensile strength; (**b**) 28 days split tensile strength.

**Figure 11 materials-15-07157-f011:**
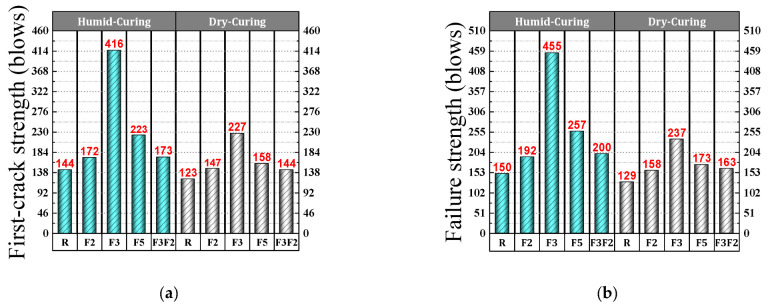
Average blows for concrete discs: (**a**) first-crack strength; (**b**) failure strength.

**Figure 12 materials-15-07157-f012:**
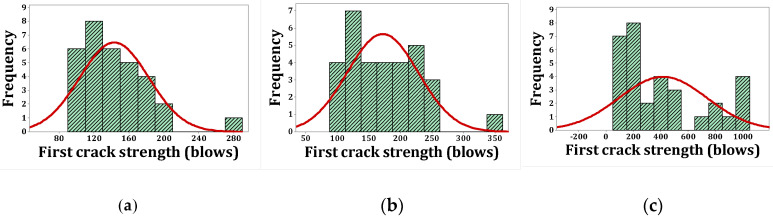
Distribution plots for first crack (humid): (**a**) R, (**b**) F2, (**c**) F3.

**Figure 13 materials-15-07157-f013:**
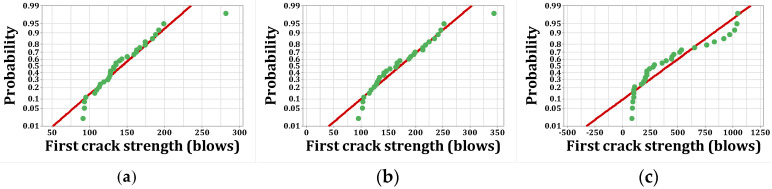
Normal probability plots for first crack (humid): (**a**) R, (**b**) F2, (**c**) F3.

**Figure 14 materials-15-07157-f014:**
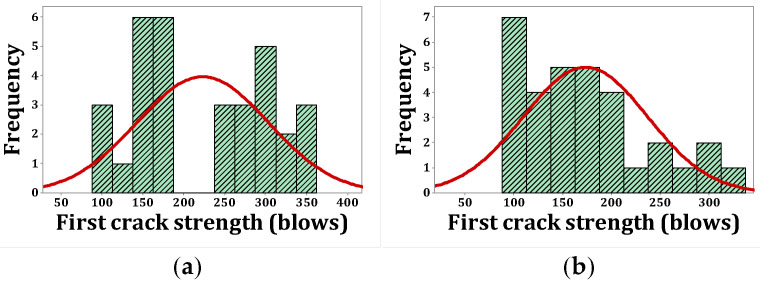
Distribution plots for first crack (humid): (**a**) F5, (**b**) F2F3.

**Figure 15 materials-15-07157-f015:**
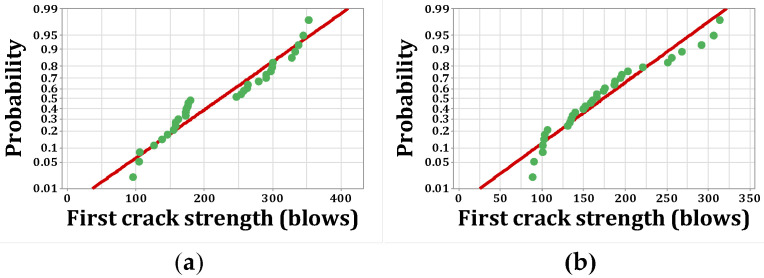
Normal probability plots for first crack (humid): (**a**) F5, (**b**) F2F3.

**Figure 16 materials-15-07157-f016:**
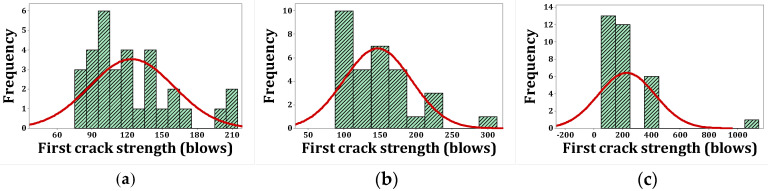
Distribution plots for first crack (dry): (**a**) R, (**b**) F2, (**c**) F3.

**Figure 17 materials-15-07157-f017:**
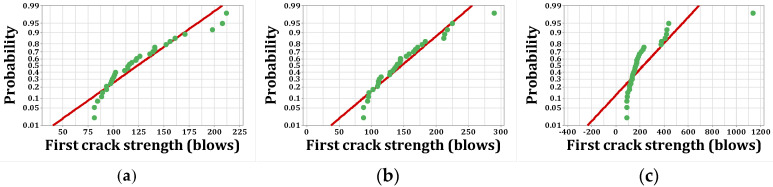
Normal probability plots for first crack (dry): (**a**) R, (**b**) F2, (**c**) F3.

**Figure 18 materials-15-07157-f018:**
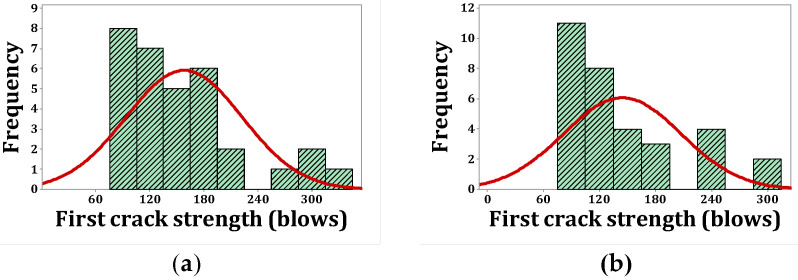
Distribution plots for first crack (dry): (**a**) F5, (**b**) F2F3.

**Figure 19 materials-15-07157-f019:**
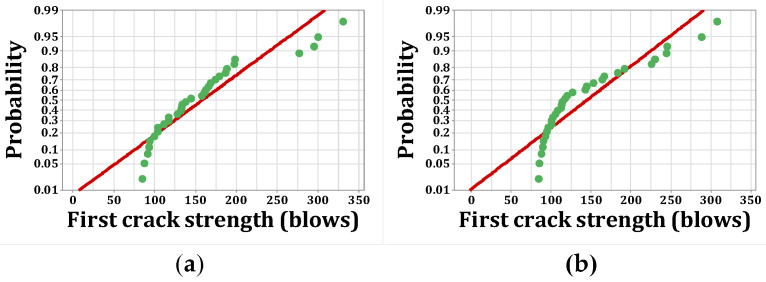
Normal probability plots for first crack (dry): (**a**) F5, (**b**) F2F3.

**Figure 20 materials-15-07157-f020:**
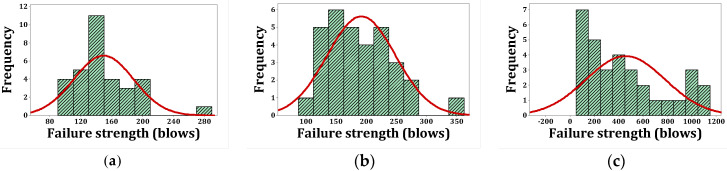
Distribution plots for failure strength (humid): (**a**) R, (**b**) F2, (**c**) F3.

**Figure 21 materials-15-07157-f021:**
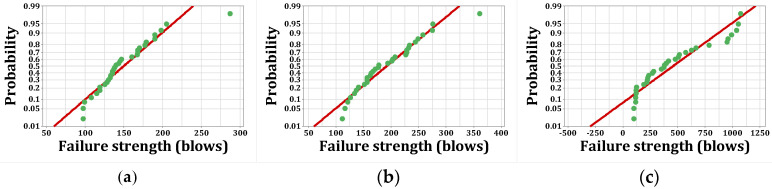
Normal probability plots for failure strength (humid): (**a**) R, (**b**) F2, (**c**) F3.

**Figure 22 materials-15-07157-f022:**
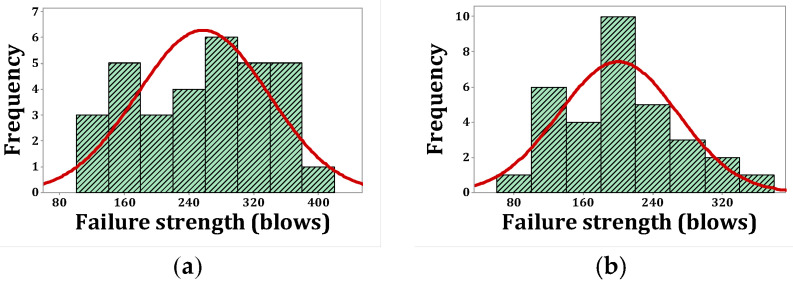
Distribution plots for failure strength (humid): (**a**) F5, (**b**) F2F3.

**Figure 23 materials-15-07157-f023:**
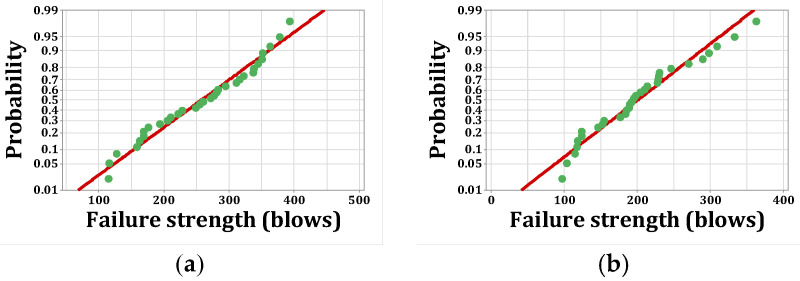
Normal probability plots for failure strength (humid): (**a**) F5, (**b**) F2F3.

**Figure 24 materials-15-07157-f024:**
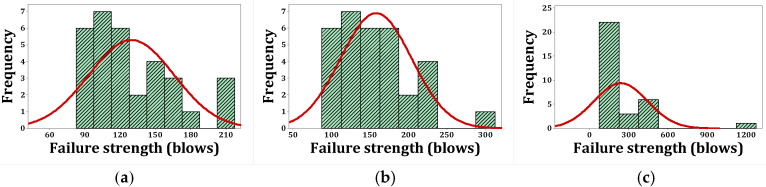
Distribution plots for failure strength (dry): (**a**) R, (**b**) F2, (**c**) F3.

**Figure 25 materials-15-07157-f025:**
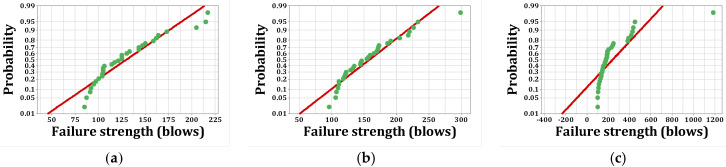
Normal probability plots for failure strength (dry): (**a**) R, (**b**) F2, (**c**) F3.

**Figure 26 materials-15-07157-f026:**
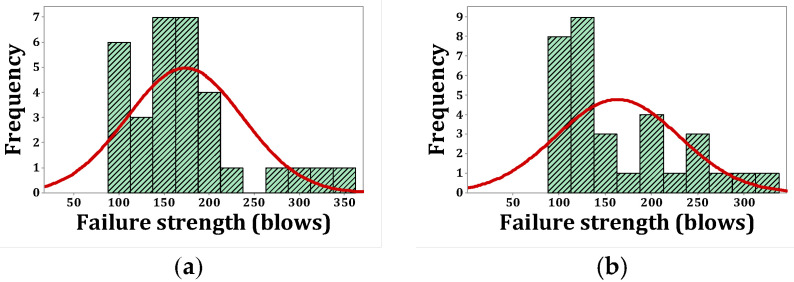
Distribution plots for failure strength (dry): (**a**) F5, (**b**) F2F3.

**Figure 27 materials-15-07157-f027:**
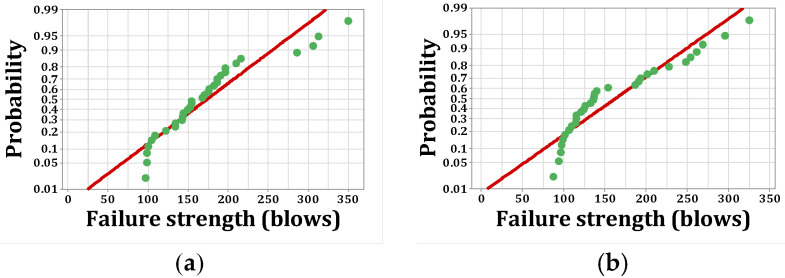
Normal probability plots for failure strength (dry): (**a**) F5, (**b**) F2F3.

**Figure 28 materials-15-07157-f028:**
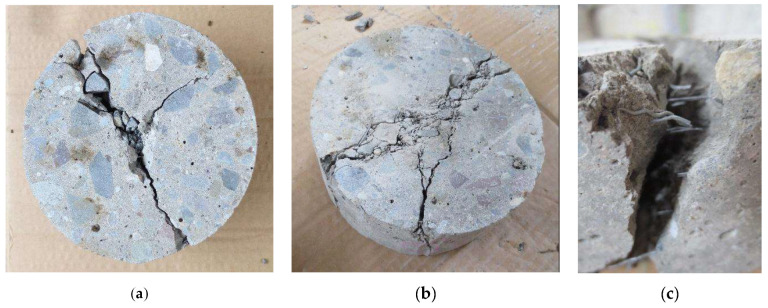
Specimens after impact test: (**a**) plain concrete, (**b**) steel fiber reinforced concrete, (**c**) fiber bridging.

**Figure 29 materials-15-07157-f029:**
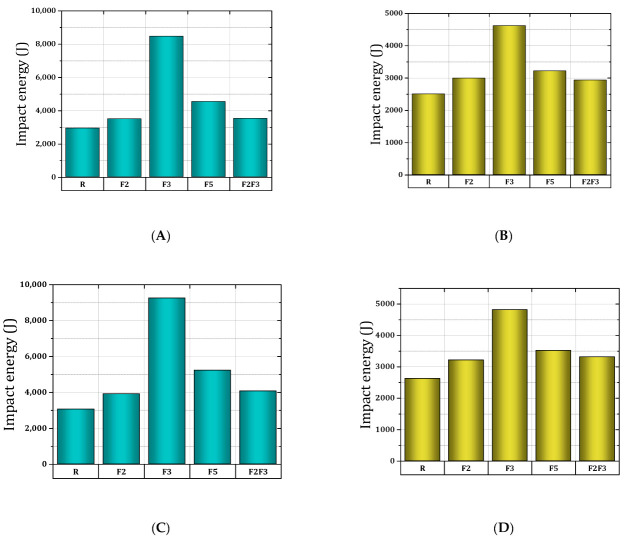
Impact energy results for first crack ((**A**): humid, (**B**): dry) and failure strength ((**C**): humid, (**D**): dry).

**Table 1 materials-15-07157-t001:** Information on previous studies.

Reference	Type of Concrete	Fiber Shape	Fiber Percentage (%)	Fiber Geometry	Mechanical Properties
*d* (mm)	*l* (mm)	*l*/*d*
Güneyisi et al. [22]	Normal + MK + SF	Hooked-end	0.25, 0.75	0.75	30, 60	40, 80	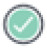 CS	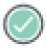 TS	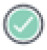 FS	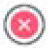 IS
Chen et al. [23]	Recycled aggregate concrete + SF	Crimped	0.5, 1, 1.5	0.8	32	40	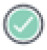 CS	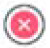 TS	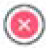 FS	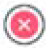 IS
Nataraja et al. [24]	Normal + SF	Crimped	0.5, 1, 1.5	1	40	40	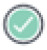 CS	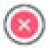 TS	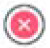 FS	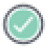 IS
Nataraja et al. [25]	Normal + SF	Crimped	0.5	0.5	27.5	55	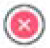 CS	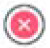 TS	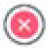 FS	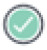 IS
Song et al. [26]	High-strength concrete + SF	Hooked-end	1	0.8	35	40	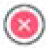 CS	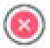 TS	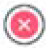 FS	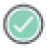 IS
Olivito and Zuccarello[27]	Normal + SF	Undefined	1, 2	0.44, 0.60, 0.8	22, 30, 44	50	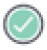 CS	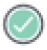 TS	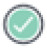 FS	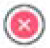 IS
Gesoglu et al. [28]	Normal + silica fume +SF	Hooked-end	0.25, 0.75	0.75	30, 60	40, 80	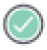 CS	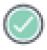 TS	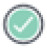 FS	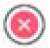 IS
Yoo et al. [29]	Textile-reinforced concrete + SF	Hooked-end	0.5, 1, 1.5, 2	0.5	30	60	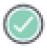 CS	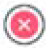 TS	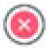 FS	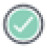 IS
Rizzuti and Bencardino[30]	Normal + SF	Hooked-end	1, 1.6, 3, 5	0.55	22	40	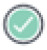 CS	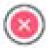 TS	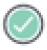 FS	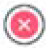 IS
Nili and Afroughsabet[19]	Normal+ silica fume +SF	Hooked-end	0.5, 1	0.75	60	80	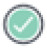 CS	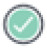 TS	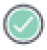 FS	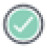 IS
Nili and Afroughsabet[31]	Normal+ silica fume +SF	Hooked-end	0.5, 1	0.75	60	80	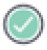 CS	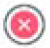 TS	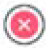 FS	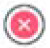 IS
Nili et al. [32]	Normal+ silica fume +SF	Hooked-end	1	0.75	50	67	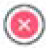 CS	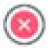 TS	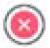 FS	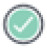 IS
Yan et al. [33]	High-strength concrete + SF	Straight	1.5	0.417	25	60	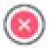 CS	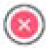 TS	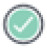 FS	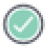 IS
Wang [34]	Lightweight aggregate concrete + SF	Slightly enlarged ends	0.5, 1, 1.5, 2	64	32	50	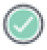 CS	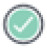 TS	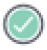 FS	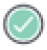 IS
Atis and Karahan [35]	Normal + fly ash + SF	Hooked-end	0, 0.25, 0.5, 1, 1.5	0.55	35	64	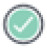 CS	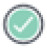 TS	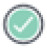 FS	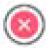 IS
Nazarimofrad et al. [36]	Recycled aggregate concrete + SF	Hooked-end	1	0.85	50	60	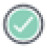 CS	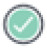 TS	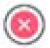 FS	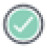 IS
Huang and Zhao [37]	Normal + SF	Undefined	1, 2	0.58	25, 35, 45	43, 60, 77	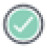 CS	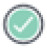 TS	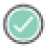 FS	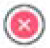 IS
Altun et al. [38]	Normal + SF	Hooked-end	0.5, 1, 1.5	0.75	60	80	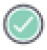 CS	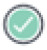 TS	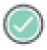 FS	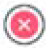 IS
Rahmani et al. [21]	Normal + SF	Hooked-end	0.5	0.55	35	64	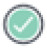 CS	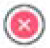 TS	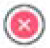 FS	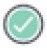 IS
Mohammadi et al. [39]	Normal + SF	Undefined	1, 1.5, 2	1.25	25, 50	20, 40	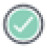 CS	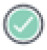 TS	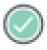 FS	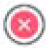 IS
Abbass et al. [17]	Normal + SF	Hooked-end	0.5, 1, 1.5	0.62, 0.75	40, 50, 60	65, 80	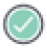 CS	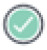 TS	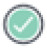 FS	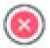 IS
Mahakavi and Chithra [40]	Self-compacting concrete + SF	Hooked-end	0.25, 0.5, 0.75	0.7	70	100	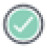 CS	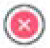 TS	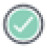 FS	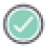 IS
Crimped	0.25, 0.5
Carneiro et al. [41]	Recycled aggregate concrete + SF	Hooked-end	0.75	0.55	35	65	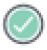 CS	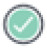 TS	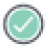 FS	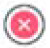 IS
Afroughsabet et al. [42]	High-performance recycled aggregate concrete + SF	Hooked-end	1	0.9	60	67	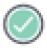 CS	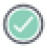 TS	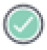 FS	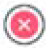 IS
Ibrahim and Bakar [43]	Normal + SF	Hooked-end	0.5, 0.75, 1, 1.25	0.75	60	80	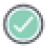 CS	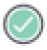 TS	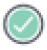 FS	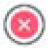 IS
Huo and Zhang [44]	Steel fiber foamed concrete	Crimped	1		48	24	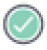 CS	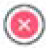 TS	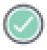 FS	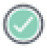 IS
Eren and Marar [45]	Limestone crusher dust and steel fibers on concrete	Hooked-end	0.5, 1, 1.5	0.50, 0.75, 0.90	50, 60	65, 80, 100	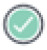 CS	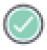 TS	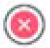 FS	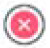 IS
Düzgün et al. [46]	Lightweight concrete + SF	Hooked-end	0.5, 1, 1.5	0.8	60	75	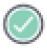 CS	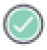 TS	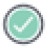 FS	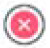 IS
Alrawashdeh and Eren [47]	Self-compacting concrete + SF	Hooked-end	0.35, 0.45, 0.55	0.5	30	60	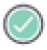 CS	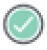 TS	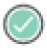 FS	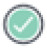 IS
0.625	50	80
Kachouh et al. [48]	Recycled aggregate concrete + SF	Hooked-end	1, 2, 3	0.55	35	65	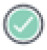 CS	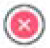 TS	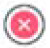 FS	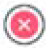 IS
Rai and Singh [49]	Normal + SF	Crimped	0.1, 0.2, 0.3 0.4, 0.5, 0.6, 0.7, 0.8, 0.9, 1, 1.25, 1.5	0.6	30	50	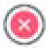 CS	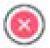 TS	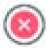 FS	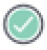 IS
Ghorpade and Rao [50]	Recycled aggregate concrete + SF	Undefined	0.5, 0.75, 1, 1.25	1	100	100	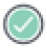 CS	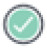 TS	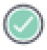 FS	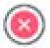 IS
Alavi Nia et al. [13]	Normal + SF	Hooked-end	0.5, 1	0.75	60	80	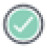 CS	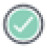 TS	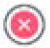 FS	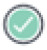 IS
Lee et al. [51]	High-strength concrete + SF	Hooked-end	0.5, 1, 1.5, 2	0.38, 0.55, 1.05	30, 35, 50	79, 64, 48	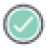 CS	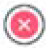 TS	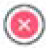 FS	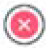 IS
Bywalski et al. [52]	High-strength concrete + SF	Straight	1, 2, 3	0.2	13	65	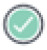 CS	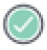 TS	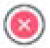 FS	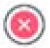 IS
Pham et al. [53]	Geopolymer concrete + SF	Hooked-end	0.5, 1, 1.5	0.5	30	60	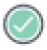 CS	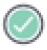 TS	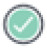 FS	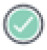 IS
Mastali et al. [54]	Self-consolidating concrete + SF	Hooked-end	0.5, 0.75, 1, 1.5	---	---	47	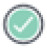 CS	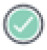 TS	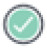 FS	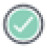 IS
Hu et al. [55]	Normal + SF	Wavy profile	0.38, 0.45, 0.57	0.8, 1.0	55, 60	69, 60	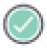 CS	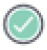 TS	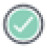 FS	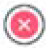 IS
Gao and Zhang [56]	Recycled aggregate concrete + SF	Hooked-end	0.5, 1, 1.5, 2	0.6	30.5	54.6	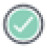 CS	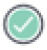 TS	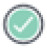 FS	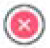 IS
Ismail et al. [57]	Self-consolidating rubberized concrete + SF	Hooked-end	0.35, 0.5, 0.75, 1	0.55	35	65	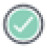 CS	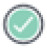 TS	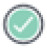 FS	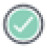 IS
0.90	60	65
Gao et al. [58]	Recycled aggregate concrete + SF	Hooked-end	0.5, 1, 1.5, 2	0.6	30.5	54.6	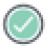 CS	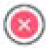 TS	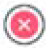 FS	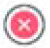 IS
Xie et al. [59]	Recycled aggregate concrete + SF	Hooked-end	1	0.2	13	65	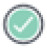 CS	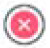 TS	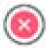 FS	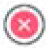 IS
Soylev and Ozturan [60]	Normal + SF	Hooked-end	0.5	0.55	35	64	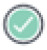 CS	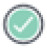 TS	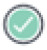 FS	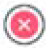 IS
Chen et al. [61]	Normal + SF	Hooked-end	0.5, 1	0.55	35	64	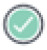 CS	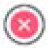 TS	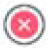 FS	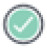 IS
Ou et al. [62]	Normal + SF	Hooked-end	0.8, 1.6, 2, 2.4, 2.6, 3, 3.4	0.5, 0.6, 1.0	30, 50, 60	50, 60, 70, 100	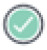 CS	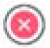 TS	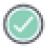 FS	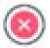 IS
Murali et al. [63]	Functionally graded concrete + SF	Hooked-end crimped	0.8–3.6	1	50	50	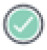 CS	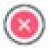 TS	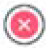 FS	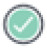 IS
Ameri et al. [64]	High-strength concrete + SF	Hooked-end	1	0.55	30	55	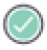 CS	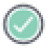 TS	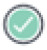 FS	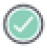 IS
Koushkbaghi et al. [65]	Recycled aggregate concrete + SF	Hooked-end	1.5	0.75	50	67	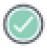 CS	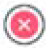 TS	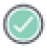 FS	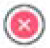 IS
Yazıcı et al. [66]	Normal + SF	Hooked-end	0.5, 1, 1.5	0.62, 0.90, 0.75	30, 60, 60	45, 65, 80	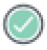 CS	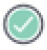 TS	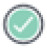 FS	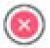 IS
Reddy and Rao [67]	Normal + SF	Crimped	0.5, 1, 1.5, 2	0.5	30	60	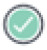 CS	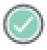 TS	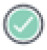 FS	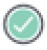 IS
Jian-he et al. [68]	Recycled aggregate concrete + SF + rubber crumb	Crimped	1	0.70	32	45	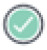 CS	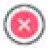 TS	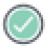 FS	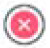 IS
Abdallah et al. [69]	Self-compacting concrete + SF	Single hooked	0.5, 1	0.90	60	67	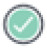 CS	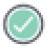 TS	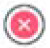 FS	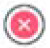 IS
Double hooked
Triple hooked
Soutsos et al. [70]	Normal + SF	Hooked-end	0.5, 1, 1.5	0.9	60	67	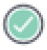 CS	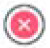 TS	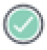 FS	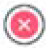 IS
Wavy profile	0.1	50, 60	50, 60
Flattened end	0.1	50	50

CS: compressive strength; TS: tensile strength; FS: flexural strength; IS: impact strength.

**Table 2 materials-15-07157-t002:** Chemical composition of cement and silica fume.

Chemical Analysis	Cement (wt%)	Silica Fume (wt%)
SiO_2_	20.23	91.44
Al_2_O_3_	4.86	1
Fe_2_O_3_	4.52	0.9
CaO	63.86	1.69
MgO	2.14	1.78
SO_3_	2.12	0.77
K_2_O	0.71	0.07
Na_2_O	0.21	0.05
LOI	0.85	2.01

**Table 3 materials-15-07157-t003:** Properties of superplasticizer.

Chemical Base	Physical State	Chlorine Ion	Color	Specific Weight
Polycarboxylates	Liquid	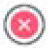	Light brown	120 ± 0.05 g/cm^3^

**Table 4 materials-15-07157-t004:** Properties of steel fibers.

Length (mm)	Diameter (mm)	Aspect Ratio (l/d)	Specific Gravity	Tensile Strength (MPa)	Geometry
20	0.75	27	7.8	1140	Crimped
30	0.75	40	7.8	1140	Hooked
50	0.75	67	7.8	1140	Crimped

**Table 5 materials-15-07157-t005:** Mixed proportions of concrete.

Mix	Cement (kg.m^−3^)	SF (kg.m^−3^)	Water (kg.m^−3^)	Gravel (kg.m^−3^)	Sand (kg.m^−3^)	Steel Fibers (%)		
						Crimped Fiber	Hooked-End Fiber	Crimped Fiber
						20 mm	30 mm	50 mm
R	405	45	180	900	970	-	-	-
F2	405	45	180	900	970	1	-	-
F3	405	45	180	900	970	-	1	-
F5	405	45	180	900	970	-	-	1
F2F3	405	45	180	900	970	0.5	0.5	-

**Table 6 materials-15-07157-t006:** Characteristics of tests, specimens, and standards.

Hardened Concrete	Fresh Concrete
Test	Dimension	Standard	Shape	Test	Standard
Compressive strength	150 × 150 × 150 mm	ASTM C39-03 [75]	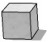	Slump flow time	BS EN 12350-2 [76]
Tensile strength	300 × 150 mm	ASTM C496-04 [77]	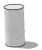		
Impact strength	300 × 150 mm	ACI committee 544 [20]	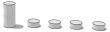

**Table 7 materials-15-07157-t007:** Repeatability test results for previous studies and the present study.

Ref.	TEST	RI
Mastali and Dalvand [78]	First crack strength	38.45
Fakharifar et al. [79]		43.66
Rahmani et al. [21]		57.5
Dalvand et al. [80]		27.56
Badr et al. [81]		59
Present Work		44
Mastali and Dalvand [78]	Failure strength	37.3
Fakharifar et al. [79]		43.66
Rahmani et al. [21]		48
Dalvand et al. [80]		26.41
Badr et al. [81]		50.4
Present Work		41.4
Mastali and Dalvand [78]	INPB	59
Fakharifar et al. [79]		49.66
Dalvand et al. [80]		46.04
Present Work		58
Mastali and Dalvand [78]	Compressive strength	6.18
Fakharifar et al. [79]		6.73
Dalvand et al. [80]		4.37
Badr et al. [81]		7.55
Present Work		3.66
Dalvand et al. [80]	Tensile strength	8.19
Present Work		7.57
Lee et al. [51]	Slump	25.60
Present Work		11.08

**Table 8 materials-15-07157-t008:** Statistical variables for initial crack strength, fracture strength, and samples.

Mix	Curing	Strength	Mean (Blows)	Standard Deviation (Blows)	Coefficient of Variation (%)	Standard Error of Mean (Blows)	95% Confidence Interval		*p*-Value of K–S Test
							Upper Bound (Blows)	Lower Bound (Blows)	
R	H U M I D	First Crack	144	40	27	7	158	130	0.093
		Failure	150	39	26	7	164	137	0.068
		INPB	6	3	45	0.515	8	4	<0.01
F2		First Crack	172	56	33	9.98	191	152	>0.15
		Failure	192	57	29	10	211	172	>0.15
		INPB	20	8	41	1.45	23	17	>0.15
F3		First Crack	416	321	77	56.8	527	305	<0.01
		Failure	455	324	71	57.3	567	342	0.087
		INPB	39	33	86	5.84	50	27	<0.01
F5		First Crack	223	81	36	14.2	251	195	<0.01
		Failure	257	81	32	14.4	285	229	>0.15
		INPB	34	23	68	4.1	42	26	0.031
F2F3		First Crack	173	64	37	11.3	195	151	>0.15
		Failure	200	69	34	12.1	224	176	>0.15
		INPB	27	16	60	2.83	32	21	0.024
R	D R Y	First Crack	123	36	29	6.39	136	111	0.141
		Failure	129	36	28	6.41	142	117	0.075
		INPB	6	2	36	0.353	6	4	<0.01
F2		First Crack	147	47	32	8.3	163	131	0.115
		Failure	158	46	30	8.19	174	142	>0.15
		INPB	11	6	49	0.971	13	9	<0.01
F3		First Crack	227	199	88	35.1	296	158	<0.01
		Failure	237	204	86	36.1	309	168	<0.01
		INPB	11	8	71	1.43	14	8	<0.01
F5		First Crack	158	65	41	11.5	180	135	0.093
		Failure	173	64	37	11.4	195	151	0.022
		INPB	15	9	56	1.5	18	12	<0.01
F2F3		First Crack	144	63	44	11.1	166	123	<0.01
		Failure	163	67	41	11.8	186	140	<0.01
		INPB	19	13	71	2.33	23	14	0.048

**Table 9 materials-15-07157-t009:** Comparison of the impact strength of FRC versus plain concrete.

Parameter		HUMID				DRY			
		F2	F3	F5	F2F3	F2	F3	F5	F2F3
*p*-Value of Kruskal–Wallis test	First Crack	0.029	0.000	0.000	0.049	0.030	0.001	0.021	0.321
	Failure	0.001	0.000	0.000	0.001	0.003	0.000	0.002	0.058
	INPB	0.000	0.000	0.000	0.000	0.000	0.000	0.000	0.000
First-crack strength of FRC/First crack of plain concrete		1.194	2.888	1.548	1.201	1.195	1.845	1.284	1.170
Failure strength of FRC/First crack of plain concrete		1.28	3.033	1.713	1.333	1.224	1.844	1.341	1.263
INPB of FRC/INPB of plain concrete		3.333	6.5	5.666	4.5	1.833	1.833	2.5	3.166

**Table 10 materials-15-07157-t010:** The results of impact energy and the ductility index.

Mix	Curing	Impact Energy (J)		Ductility Index λ
First Crack	Failure
R	H U M I D	2930.4	3052.5	0.042
F2		3500.2	3907.2	0.116
F3		8465.6	9259.25	0.094
F5		4538.05	5229.95	0.152
F2F3		3520.55	4070	0.156
R	D R Y	2503.05	2625.15	0.049
F2		2991.45	3215.3	0.075
F3		4619.45	4822.95	0.044
F5		3215.3	3520.55	0.095
F2F3		2930.4	3317.05	0.132

**Table 11 materials-15-07157-t011:** The estimation equations formulated for fracture strength.

Mix	Equations	R^2^	Mix	Equations	R^2^
R (Humid)	$N_{p-fail}=0.9752\;N_{first}+9.890\;\;$	994	R(Dry)	$N_{p-fail}=1.002\;N_{first}+5.313$	0.996
F2 (Humid)	$N_{p-fail}=0.9952\;N_{first}+20.89\;\;$	0.978	F2 (Dry)	$N_{p-fail}=0.9800\;N_{first}+14.15\;\;$	0.984
F3 (Humid)	$N_{p-fail}=1.004\;N_{first}+36.76\;\;$	0.988	F3 (Dry)	$N_{p-fail}=1.028\;N_{first}+5.106$	0.998
F5 (Humid)	$N_{p-fail}=0.9703\;N_{first}+40.69\;$	0.919	F5 (Dry)	$N_{p-fail}=0.9827\;N_{first}+17.89$	0.982
F2F3 (Humid)	$N_{p-fail}=1.048\;N_{first}+18.61$	0.946	F2F3 (Dry)	$N_{p-fail}=1.038\;N_{first}+12.99$	0.855

**Table 12 materials-15-07157-t012:** Efficiency of linear patterns for various types of fiber.

Accuracy Measures	Humid					Dry				
	R	F2	F3	F5	F2F3	R	F2	F3	F5	F2F3
*MAD*	2.264	6.669	25.876	18.434	11.758	1.736	4.548	4.894	6.312	9.578
*MAPE* (%)	1.674	3.959	8.831	7.899	6.245	1.410	3.118	2.313	4.387	6.639
R^2^	0.994	0.978	0.988	0.919	0.946	0.996	0.984	0.998	0.982	0.855

**Table 13 materials-15-07157-t013:** Output parameters and process of experiments in previous studies.

Ref.	Type of Fiber	Fiber (%)	CS	TS	IS			Proposed Equations	*MAD*	*MAPE*	R^2^	*p*-Value of K–S Test	
					FC	UC	INPB					FC	UC
Mastali and Dalvand [78]	RC	0.25	▲	▲	43	52	9	$N_{p-fail}=1.126\;N_{first}+3.3614$	-	-	0.9082	0.999	0.429
		0.75	▲	▲	60	77	17	$N_{p-fail}=1.2669\;N_{first}+0.71$	-	-	0.9426	0.999	0.512
		1.25	▲	▲	79	104	25	$N_{p-fail}=1.1872\;N_{first}+9.9145$	-	-	0.269	0.684	0.999
Fakharifar et al. [79]	PP	0.5	▲	-	40	48	8	$N_{p-fail}=1.27\;N_{first}-1.01$	-	-	0.94	>0.15	>0.15
		0.75	▲		52	68	16	$N_{p-fail}=1.27\;N_{first}+1.68$	-	-	0.93	>0.15	>0.15
		1	▲		62	81	19	$N_{p-fail}=1.23\;N_{first}+5.27$	-	-	0.94	>0.15	>0.15
Mastali et al. [96]	RG	0.25	▲	-	38	47	9	$N_{p-fail}=1.258\;N_{first}-0.722$	-	-	0.965	0.999	0.271
		0.75	▲		56	71	15	$N_{p-fail}=1.205\;N_{first}+3.5999$	-	-	0.989	0.999	0.834
		1.25	▲		76	98	22	$N_{p-fail}=1.3043\;N_{first}-0.7664$	-	-	0.991	0.685	0.568
Rahmani et al. [21]	C	0.15	▲	-	112	118	6	$N_{p-fail}=1.01\;N_{first}+5.17$	2.522	2.261	0.996	>0.15	>0.15
	PP	0.15	▲		56	71	15	$N_{p-fail}=1.03\;N_{first}+12.96$	3.737	6.905	0.984	0.012	<0.010
	S	0.5	▲		123	228	105	$N_{p-fail}=1.60\;N_{first}$	81.79	37.496	0.844	0.073	0.036
Song et al. [97]	S	0.5	▲	-	234	330	96	$N_{p-fail}=1.189\;N_{first}+51.569$	44.36	-	0.915	0.000	0.000
	S + PP	0.5 + 0.1	▲	-	247	356	109	$N_{p-fail}=1.008\;N_{first}+106.887$	34.73	-	0.843	0.01	0.001
Rai and Singh [49]	PP	0.05–0.5	-	-	32	52	7	$N_{p-fail}=1.7667\;N_{first}-3.4025$	19.86	7.34	0.918	>0.05	>0.05
	S	0.1–1.5	-	-	97	191	94	$N_{p-fail}=1.8479\;N_{first}+12.2438$	34.42	20.53	0.827	0.042	0.044
	S + PP	[0.2–0.5] + [0.2–0.5]	-	-	137	267	130	$N_{p-fail}=1.986\;N_{first}-7.2763$	2.17	5.22	0.987	0.040	0.048
Song et al. [26]	S	1	▲	-	1734	1896	162	$N_{p-fail}=1.031\;N_{first}+107.254$	-	-	0.980	next to zero	next to zero

FC: first crack, UC: ultimate crack, RC: recycled carbon, PP: polypropylene, RG: recycled glass, C: cellulose, S: steel.

## Data Availability

Not applicable.

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
