# Peer review of "Investigation of Impact Resistance of High-Strength Portland Cement Concrete Containing Steel Fibers"

_materials, 2022, doi:10.3390/ma15207157_

Round 1

Reviewer 1 Report

The authors performed three types of tests on steel fiber reinforced concrete materials, namely compression tests, indirect tensile strength test (splitting test), and impact resistance tests. The manuscript can be accepted once the following issues are solved:

1. In this paper, the authors use the first-point of view word i.e., "We". For the academic writing, the authors should use a passive sentence instead of using the first-or-third point of view words.

2. The authors collected test results of previous researchers. It seems that those results are not used to compare the test results obtained by authors. It is suggested if the authors could use the previous test results in test results analysis. For example: Do the previous test results present the similar trend as the current test results do? 

Author Response

Dear reviewer

I hope all is well with you.

First, thank you for your consideration.  I have attached the Word File. Please see the attachment

Kind Regards

Reviewer 2 Report

REVIEW OF RESEARCH MANUSCRIPT:

 Investigation of Impact Resistance of High-Strength Portland 2 Cement Concrete Containing Steel Fibers

The manuscript under review is related to work carried out on Investigation of Impact Resistance of High-Strength Portland 2 Cement Concrete Containing Steel Fibers ; it is experimental study that also discusses lot of experimental data from the different researcher and show the extension of the existing research through their own testing, emphasizing on FIBER additions in cement. Literature review provided is adequate.

The significance of the work is high, as, thanks to the developed testing protocol and lot of experimental data, which was available and obtained. In this work, an original set of data is provided, that clearly relates with the title provided. The quality of the data is good, the work is of great scientific quality. Most part of the writing and the figures are also clear, however partly it is required to be improved. The article overall is written good, and authors describe the content of the article upto a great extent. In short, the work is considered as particularly good and linked to domain of this journal. However, some minor questions and comments follow for the clarification and improvement of the manuscript.

1.       Properties of all individual materials may be added in research paper, fibers may also be tested to report its properties.

2.       Analytical expression may also be added for computation of potential energy released by impact test, see relevant articles.

3.       In additional to statistical expressions, analytical expression can also be reported for impact strength. Similarly, results can be presented not only for No of blows but also if form of energy absorbed.

4.       Reasoning to validate the results may be provided in details and may be validated through existing articles.

5.       Recommendations for implementation of this work practically may be added.

6.       In Table 5, the 96% Confidence Interval (CI) is given but there is no mention of this interval in the body text of the manuscript. It is suggested to describe it in a little detail so that the readers would have some idea regarding what the authors try to convey through 95 CI and its upper and lower bound.

7.       What does the acronyms exactly stand for, MAD and MAPE?

8.       MAD must have some units (for instance in terms of strength in this case, may be) which are not shown in the paper.

9.       In the equations for MAD and MAPE, there is given “n” whereas in the succeeding paragraph (line 376), the authors refer to “N” which must be replaced by “n”.

10.   Authors must cite any existing literature from where the two equations ‘MAD’ and ‘MAPE’ have been taken.

11.   In line 378, the authors illustrated that all mixtures exhibit low values of MAD and MAPE. Now that they are talking about MAD, how can they claim that these values are low unless they compare with some other values (may be of strength)?

12.   Generally, the correlation coefficient is represented by R or r (also called Pearson correlation coefficient), the authors hereby are confusing the correlation coefficient R with the coefficient of determination R2 (looking at line 364, Table 7, and line 367-368). Please remove the inconsistency that lies in the description as well as in Table 7. I mean either clearly denote the values by R2 in the Table 7 instead of R or replace R2 by R in line 367-368.

Author Response

Dear reviewer

I hope all is well with you.

First, thank you for your consideration.  I have attached the Word File. Please see the attachment.

Kind Regards

Reviewer 3 Report

Comments:

--Review of: “Investigation of Impact Resistance of High-Strength Portland Cement Concrete Containing Steel Fibers” by Mohammad Mohtasham Moein et al. The experiments of this manuscript have been carried out well and some questions are below to help you improve the manuscript.

--Fig.1 reported the characteristics of steel fibers and the types of tests. The information in the table is well and in details while its summarize about the table is less. The summarize should tell us the shortcomings of the previous studies.

--Line 81:"...the study of this type of concrete against impact load due to high dispersion of the results still needs further investigation..." Here the research-gap of this work should be summarized more specific not only the high dispersion.

--Line 98: More information of the raw materials should be listed here, such as chemical information of cement and silica fume.

--In figure 4: Slump comparisons of the five concrete should be summarized in details here.

--In 3.3. Tests: the testing parameters of experiments should be performed in details here for the repeatability test.

--In Figure 10, the F5 fiber has the most positive effect on the concrete compressive strength while the most positive effect on the concrete splitting tensile strength is the F3. How about the connection between the two aspects?

--In this work, different curing environments has been considered as the influence factor on concrete performance. Generally, the humid curing environment results in higher concrete performance compared to the dry curing environment. The reason is attributed to the higher degree of hydration of the former. More tests or literature can be executed for this point.

Author Response

(The authors gave the same response as above.)
